# PCAS: An Integrated Tool for Multi-Dimensional Cancer Research Utilizing Clinical Proteomic Tumor Analysis Consortium Data

**DOI:** 10.3390/ijms25126690

**Published:** 2024-06-18

**Authors:** Jin Wang, Xiangrong Song, Meidan Wei, Lexin Qin, Qingyun Zhu, Shujie Wang, Tingting Liang, Wentao Hu, Xinyu Zhu, Jianxiang Li

**Affiliations:** School of Public Health, Suzhou Medical College of Soochow University, Suzhou 215123, China; jinwang93@suda.edu.cn (J.W.); 20234247003@stu.suda.edu.cn (X.S.); 20234247035@stu.suda.edu.cn (M.W.); 2330506055@stu.suda.edu.cn (L.Q.); 2230504071@stu.suda.edu.cn (Q.Z.); 2230412065@stu.suda.edu.cn (S.W.); 2230412060@stu.suda.edu.cn (T.L.); 2330506024@stu.suda.edu.cn (W.H.); 2330506091@stu.suda.edu.cn (X.Z.)

**Keywords:** proteomics, phosphoproteomics, clinical proteomic tumor analysis consortium (CPTAC), integrated cancer analysis, R package, shiny application

## Abstract

Proteomics offers a robust method for quantifying proteins and elucidating their roles in cellular functions, surpassing the insights provided by transcriptomics. The Clinical Proteomic Tumor Analysis Consortium database, enriched with comprehensive cancer proteomics data including phosphorylation and ubiquitination profiles, alongside transcriptomics data from the Genomic Data Commons, allow for integrative molecular studies of cancer. The ProteoCancer Analysis Suite (PCAS), our newly developed R package and Shinyapp, leverages these resources to facilitate in-depth analyses of proteomics, phosphoproteomics, and transcriptomics, enhancing our understanding of the tumor microenvironment through features like immune infiltration and drug sensitivity analysis. This tool aids in identifying critical signaling pathways and therapeutic targets, particularly through its detailed phosphoproteomic analysis. To demonstrate the functionality of the PCAS, we conducted an analysis of GAPDH across multiple cancer types, revealing a significant upregulation of protein levels, which is consistent with its important biological and clinical significance in tumors, as indicated in our prior research. Further experiments were used to validate the findings performed using the tool. In conclusion, the PCAS is a powerful and valuable tool for conducting comprehensive proteomic analyses, significantly enhancing our ability to uncover oncogenic mechanisms and identify potential therapeutic targets in cancer research.

## 1. Introduction

Proteomics is a powerful method for studying the composition and function of proteins in cells, tissues, and organisms [1,2]. Tandem Mass Tag (TMT) technology, a mass spectrometry-based quantitative proteomics technique, enables the relative quantification of proteins by labeling peptides from different samples with specific chemical tags. These labeled peptides are then mixed and analyzed using mass spectrometry, resulting in the identification and quantification of proteins in different samples [3]. Fluctuations of gene expression in lung cancer progress can be measured by analyzing transcriptome or proteome levels. Microarray and sequencing technologies have been developed for the large-scale analysis of the transcriptome. According to central dogma, there is a correlation between the cellular levels of proteins and their corresponding mRNAs. However, it has been reported that only ∼40% of the change in protein levels can be interpreted by measuring mRNA levels, and the remaining alterations may be due to the presence of various post-transcriptional regulation mechanisms [4,5].

The Clinical Proteomic Tumor Analysis Consortium (CPTAC, https://proteomics.cancer.gov/programs/cptac, access data: 27 March 2024) database, supported by the National Cancer Institute (NCI), is a valuable resource for cancer research. It contains extensive proteomics data from cancer samples, including protein expression levels and post-translational modifications such as phosphorylation and ubiquitination [6]. Also, researchers can access transcriptomics data of CPTAC research cohorts through the Genomic Data Commons portal (https://portal.gdc.cancer.gov/, access data: 27 March 2024). Integrating the proteomics data and transcriptomics data allows for comprehensive molecular studies of cancer and its biological characteristics.

To facilitate the use of CPTAC data for researchers, we have developed the ProteoCancer Analysis Suite (PCAS). The PCAS is an R package that leverages the CPTAC database to integrate proteomics, phosphoproteomics, and transcriptomics for cancer research. Additionally, we have developed a user-friendly Shinyapp that allows for visual and interactive data analysis. This tool simplifies data analysis across multiple cancer types, enhancing the understanding of the tumor microenvironment and aiding in the development of targeted therapies. By providing both an R package and a Shinyapp (https://jingle.shinyapps.io/PCAS/, accessed on 15 April 2024), the PCAS significantly empowers researchers by making CPTAC data more accessible and enabling comprehensive analyses to uncover critical insights into cancer biology and identify potential therapeutic targets.

## 2. Results

### 2.1. Module 1: Single Dataset Analysis

#### 2.1.1. Single Gene Analysis

We will show an example of the GAPDH expression analysis using the LUAD_CPTAC_protein dataset. The dataset contains 110 tumor tissues and 101 normal tissues dissected from lung adenocarcinoma (LUAD) patients [7]. To make a visual analysis, the user has to select the following parameters on the Shinyapp, with all other parameters set to their default values.

(a) Select the Proteome and Phosphoproteome for demonstration.

(b) Lung is chosen for demonstration.

(c) The dataset interface lists all data under the selected cancer type.

(d) Select LUAD_CPTAC_protein, or choose this dataset from the ‘select dataset’ dropdown menu.

(e) Select GAPDH from the dropdown menu.

(f) The expression analysis interface is used for the visualization of the expression levels of the input gene in the selected dataset.

(g) The survival analysis interface facilitates the survival analysis and the plotting of Kaplan–Meier survival curves.

(h) The clinical data interface is used to extract clinical data from the selected dataset and merges it with protein expression data; additionally, the ‘stage plot’ sub-interface displays a bar plot of selected clinical–pathological features, and continuous variables can be set to cutoff points to be converted into categorical variables.

As shown in Figure 1B, in the LUAD_CPTAC_protein dataset, the expression of GAPDH in tumor tissue is significantly higher than in normal tissue (*p* < 0.001, Figure 1B). For the survival analysis, dividing LUAD samples into high and low expression groups at the optimal cutoff point shows that the prognosis of the low expression group is significantly better than the overexpression group (*p* < 0.001, Figure 1C). Regarding clinical data, clicking ‘Fetch clinic data’ retrieves the clinical data of the selected dataset and merges it with the expression data (Figure 1D). Furthermore, we examined the expression of the classical tumor suppressor gene PTEN in these datasets. The results showed that both the mRNA and protein expression levels of PTEN were significantly higher in tumors than in normal tissues in the LUAD_CPTAC cohort (Appendix A). At the phosphorylation level, the phosphorylation at the s294 site was significantly higher in tumor samples compared to normal tissues (Appendix A). The KM survival curve also indicated that patients with high PTEN expression had a significantly better prognosis compared to those with low expression (Appendix A).

As depicted in Figure 1E, the expression of GAPDH increases with the progression of the pathologic T stage. When selecting the LUAD_CPTAC_mRNA dataset, the mRNA levels of *GAPDH* in tumor tissues are significantly higher than in normal tissues (*p* < 0.001, Figure 2A), and the prognosis of the low expression group is significantly better than that of the high expression group when using the optimal cutoff point of *GAPDH* expression (*p* < 0.001, Figure 2B). For the phosphoproteomics data, we analyzed two phosphorylation sites of GAPDH; namely, NP_001276674.1:s210 and NP_001276674.1:s83, were used for the demonstration analysis. The results show that the phosphorylation levels at these sites in tumor tissues are significantly higher than in normal tissues (*p* < 0.001, Figure 2C,E), and patients with lower phosphorylation levels have a significantly better prognosis than those with higher levels (*p* < 0.001, Figure 2D,F).

#### 2.1.2. Multi-Gene Expression Analysis

This interface enables the differential expression 
analysis of a gene list, demonstrated here with m^6^A reader proteins YTHDC1/2, 
YTHDF1/2/3, 
and IGFBP1/2/3 in the LUAD_CPTAC_protein 
dataset.

(a) Select the data type as Proteome.

(b) Choose the cancer type as Lung.

(c) Select the LUAD_CPTAC_protein dataset.

(d) Enter the genes (proteins) to be analyzed, one at a time.

(e) Select the differential analysis algorithm, here demonstrated with the *t*-test.

After clicking the submit button, grouped box plots of these proteins are displayed. The results indicate that except for IGF2BP1/2/3, the expression of other proteins is significantly higher in tumor tissues compared to normal tissues (all *p* < 0.001, Figure 3A). When selecting the LUAD_CPTAC_mRNA dataset to analyze changes in RNA levels of these reader proteins, the results show that *IGF2BP1/3* and *YTHDF1/2* RNA levels are significantly higher in tumor tissues compared to normal tissues (all *p* < 0.001, Figure 3B), whereas *IGFBP2*, *YTHDC1/2*, and *YTHDF3* are expressed higher in normal tissues (all *p* < 0.001, Figure 3B). For phosphorylation, we analyzed the phosphorylation levels of YTHDC2 phosphorylation sites, which can be obtained from the Phoso-sites visualization module. The results indicate that the phosphorylation levels of these sites in tumor tissues are significantly higher than in normal tissues (all *p* < 0.05, Figure 3C).

#### 2.1.3. Correlation Analysis

This analysis component can examine the correlation of a target gene with multiple genes; here the correlation between GAPDH and m^6^A reader protein expression in the LUAD_CPTAC_protein dataset is analyzed. The analysis process is as follows:

(a) Select variable A data type as Proteome.

(b) Choose the cancer type as Lung.

(c) Select the LUAD_CPTAC_protein dataset.

(d) From the dropdown menu under ‘Input gene A symbol’, select GAPDH.

(e) Choose the dataset for variable B, which will automatically update to the same research cohort’s dataset according to variable A’s dataset. Here, select LUAD_CPTAC_protein, LUAD_CPTAC_mRNA, and LUAD_CPTAC_phoso; in this case, LUAD_CPTAC_protein is chosen.

(f) Select the gene list for variable B, which includes eight m^6^A reader proteins.

(g) Choose the sample types for analysis, including tumor, normal, or both, with all selected for demonstration here.

(h) Choose the correlation analysis method; here the Pearson correlation is chosen for demonstration.

(i) Clicking on data in the results table will display the correlation scatter plot and specific data for the selected row.

After submitting, the correlation analysis results are obtained shortly thereafter (Figure 4A). The results show significant correlations between GAPDH protein expression and several reader proteins, including IGF2BP1 (r = 0.358, *p* < 0.001), IGF2BP3 (r = 0.313, *p* < 0.001), IGF2BP2 (r = 0.267, *p* < 0.001), YTHDF3 (r = 0.224, *p* < 0.001), and YTHDF2 (r = 0.202, *p* < 0.001). Clicking on the row data for IGF2BP1 in the results table displays the correlation scatter plot between IGF2BP1 and GAPDH (Figure 4A). When only tumor samples are included, the correlation analysis based on tumor samples still shows significant positive correlations between GAPDH protein expression and IGF2BP1 (r = 0.455, *p* < 0.001, Figure 4B). When the data type for GAPDH is changed to transcriptome, a further correlation analysis between m^6^A reader proteins and *GAPDH* RNA levels is conducted, where IGF2BP1 shows a significant positive correlation with *GAPDH* RNA level expression (r = 0.284, *p* = 0.009, Figure 4C).

#### 2.1.4. Differential Expression Analysis

This analysis module offers a differential expression analysis for RNA and proteins based on the limma package and *t*-tests, providing comprehensive results and visualizing the data through volcano plots. We will show an example of differential protein expression using the LUAD_CPTAC_protein dataset. To perform the visual analysis, the user must select following options in the application:

(a–c) Select the data type, cancer type, and dataset.

(d) Choose the method for differential analysis, here exemplified using limma.

(e) Set the thresholds for log2 fold change (log2FC) and *p*-value for differentially expressed proteins/genes.

(f) Select genes to be marked on the volcano plot; here, the two genes with the highest log2FC in this dataset are entered for demonstration.

After clicking the Go button, the differential analysis results and volcano plot based on the limma package are obtained, identifying 73 upregulated and 451 downregulated proteins at thresholds of |log2FC| > 1 and *p* < 0.05 (Figure 5A). When using *t*-tests to identify differentially expressed proteins under the same threshold conditions, 71 upregulated and 451 downregulated proteins are found (Figure 5B). When selecting the LUAD_CPTAC_mRNA dataset for the differential RNA expression analysis, 1612 upregulated and 3214 downregulated RNAs are identified (Figure 5C).

### 2.2. Module 2: Multiple Dataset Analysis

#### 2.2.1. Gene Expression Analysis

We will show an example of GAPDH expression analysis using multiple proteomic datasets. To perform a visual analysis, the user has to select the following options on the application:

(a) Select the data type; here, we choose proteomics for demonstration.

(b) Choose the datasets to be included in the analysis; here, the default selection is datasets from the CPTAC project.

(c) Select the protein or gene for analysis.

(d) Choose the algorithm for differential analysis; here, we use the *t*-test for demonstration.

After clicking ‘Go’ to submit the analysis, the results as shown in Figure 6A indicate that GAPDH is significantly upregulated in CCRCC_CPTAC, LSCC_CPTAC, LUAD_CPTAC, PDAC_CPTAC, and UCEC_CPTAC1/2 (all *p* < 0.01), and significantly downregulated in GBM_CPTAC and HNSCC_CPTAC datasets (all *p* < 0.01). When analyzing their transcriptomic data, the results show that *GAPDH* RNA expression is significantly upregulated in all datasets except HNSCC_CPTAC (all *p* < 0.05, Figure 6B). For the phosphorylation site NP_001276674.1:s210, the analysis across multiple phosphoproteomic datasets shows that, except for in GBM_CPTAC and HNSCC_CPTAC, the phosphorylation levels in tumor samples are significantly higher than in normal samples across the other datasets (all *p* < 0.05, Figure 6C).

Additionally, *PTEN*, a tumor suppressor gene, and MKI67, a marker for cell proliferation, were also used to demonstrate the analysis across multiple datasets. As shown in Appendix A, *PTEN* mRNA and protein expression were significantly downregulated in multiple datasets, consistent with its biological role. Conversely, MKI67, as a proliferation marker, showed significantly upregulated mRNA and protein expression across various cancers (Appendix A).

#### 2.2.2. Correlation Analysis

This analysis module is used to analyze the correlation between a target gene and multiple genes across several datasets, with results visualized in a heatmap. We will show an example of the correlation analysis between GAPDH and m^6^A regulatory genes; the user has to select the following analysis modules on the application:

(a–c) As before, select the dataset and gene/protein name for variable A.

(d) Choose the data type for variable B.

(e) Select or input the gene set.

(f) Set parameters, including sample type and method for correlation analysis.

After submitting the analysis, the results and the corresponding heatmap are obtained (Figure 7A). The results show that, when including only normal samples, GAPDH exhibits significant negative correlations with several m^6^A regulatory proteins across multiple proteomic datasets. Additionally, we analyzed the correlation between GAPDH proteins and m^6^A regulatory proteins across multiple datasets including all samples using the PCAS (Figure 7B). When analyzing the correlation between *GAPDH* RNA expression and m^6^A regulatory proteins, results that were seemingly inconsistent or even opposite to the protein levels were obtained (Figure 7C). For lung cancer datasets, namely LUAD_CPTAC and LUSC_CPTAC, both GAPDH RNA and protein levels show significant positive correlations with most m^6^A regulatory proteins (*p* < 0.05, Figure 7B,C).

#### 2.2.3. Immune Cell Infiltration and Drug Sensitivity

Based on transcriptomic data, we obtained immune cell infiltration scores calculated using eight different algorithms, and drug sensitivity scores calculated using the OncoPredict package [8]. To explore the correlation between the target gene’s protein or RNA level expression and immune cell infiltration, as well as drug sensitivity, the user has to select the following analysis modules on the tool.

(a–c) As before, select the dataset and the gene/protein name.

(d) For the immune cell infiltration analysis, choose the analysis algorithm; here, we use TIMER as an example. For the drug sensitivity analysis, select the target pathway for the drug; here, we choose apoptosis regulation for demonstration.

(e) Choose the correlation analysis method.

After submitting the analysis, the results and the corresponding heatmap for the correlation analysis are obtained. For the immune cell infiltration analysis, the results indicate that GAPDH protein expression is significantly positively correlated with neutrophil infiltration scores in five datasets (*p* < 0.05, Figure 8A). Additionally, GAPDH protein expression was found to be significantly correlated with T-cell and B-cell infiltration in multiple datasets (*p* < 0.05, Figure 8A).

For the drug sensitivity analysis, the results indicate that GAPDH protein expression is significantly correlated with sensitivity to various drugs (*p* < 0.05, Figure 8B).

### 2.3. Module 3: Visualization of Protein Structure and Phosphorylation Sites

This module is used for visualizing protein domains, regions, motifs, and phosphorylation sites, which can serve as inputs for phosphorylation-related analyses. These protein structure data are sourced from the UniProt protein database; hence, this module not only provides all phosphorylation site information designed in the CPTAC database but also includes phosphorylation site data from the UniProt database. You need to set the following two parameters:

(a) Protein symbol.

(b) Select phosphorylation site data source.

For example, using the GAPDH protein, it contains a WARS1 binding region and a motif (Figure 9A). For phosphorylation sites, based on CPTAC phosphoproteomics data, a total of 27 sites are obtained (Figure 9A), while UniProt database annotations include only 16 phosphorylation sites (Figure 9B).

### 2.4. Validation: Regulation of GAPDH by IGF2BP1

To validate the reliability of the analysis results generated with this tool, we examined the m^6^A modifications on *GAPDH* RNA and its regulation by IGF2BP1. Using the sequence-based RNA adenosine methylation site predictor (SRAMP) online tool, we predicted three putative m^6^A modification sites on the mature RNA of *GAPDH* (Figure 10A). These sites were verified in two types of lung cancer cell lines using meRIP-qPCR, showing enrichment at all three sites, particularly at site 2 (Figure 10B,C). Additionally, a significant reduction in *GAPDH* mRNA levels was observed in IGF2BP1 knockdown cells (Figure 10D,E). The RNA stability assays indicated a decreased half-life of *GAPDH* RNA in cells with knocked-down IGF2BP1, further validating the regulatory effect of IGF2BP1 on *GAPDH* (Figure 10F,H).

## 3. Discussion

The analysis of proteomics data is crucial for uncovering the biology of cancer and identifying potential therapeutic targets. The CPTAC database provides valuable proteomic and phosphoproteomic data for cancer research. Platforms such as UALCAN offer pre-processed protein expression data, facilitating the comparison of protein expression differences between tumor and normal tissues [9,10]. The cProSite platform simplifies the need for bioinformatics expertise through a user-friendly interface and fast online analysis capabilities, supporting complex data comparisons and protein correlation analyses [11]. In contrast, our newly developed ProteoCancer Analysis Suite (PCAS) not only integrates these functionalities but also extends the analysis modules to include immune cell infiltration and drug sensitivity analysis, providing researchers with a comprehensive solution from single to multi-dataset integrative analyses, enhancing the understanding of the tumor microenvironment.

Protein phosphorylation is a key regulatory mechanism in cell signaling, crucial for understanding cell proliferation, differentiation, death, and other cellular functions. By analyzing phosphoproteomic data, we can identify critical signaling nodes and potential therapeutic targets in cancer [12,13]. These phosphorylation events may affect protein activity and stability, thereby regulating the behavior of tumor cells. Thus, a thorough analysis of these modifications will help us better understand the molecular mechanisms of tumor development and provide a scientific basis for developing more targeted treatment strategies. The PCAS tool offers a detailed analysis of phosphorylation sites, enabling researchers to effectively explore these critical biomarkers and assess their roles in cancer progression.

Immune cell infiltration plays a crucial role in the tumor microenvironment, where its pattern and intensity can significantly affect tumor immune evasion and therapeutic response [14,15]. By analyzing immune cell infiltration within tumor tissues, researchers can better predict patient responses to immunotherapies, such as immune checkpoint inhibitors. Furthermore, drug sensitivity analysis helps identify which tumor phenotypes are more sensitive to specific treatments, thereby guiding the formulation of personalized treatment plans [16]. Accurately predicting individual patients’ drug responses is a critical prerequisite for personalized medicine. Previous research has developed a machine learning model system that accurately predicts chemosensitivity based on proteomic approaches [17]. The PCAS tool integrates these analytical tools, not only enhancing the accuracy of predictions but also providing robust data support for clinical decisions, accelerating the translation from basic research to clinical application.

Based on our comprehensive analysis of *GAPDH* using this tool, we found that both *GAPDH* protein and RNA levels are significantly upregulated in various cancers, suggesting that *GAPDH* may act as an oncogene, consistent with our previous analyses using the TCGA database [18]. Regarding the regulatory mechanisms of *GAPDH*, our previous studies have identified its regulation in tumors through DNA methylation, copy number variations, and the transcription factor FOXM1 [18]. Other studies have shown that GAPDH plays crucial roles in tumor cell survival, angiogenesis, the regulation of gene expression in tumor cells, and the post-transcriptional regulation of tumor cell mRNA. Each of these activities is associated with increased tumor progression [19]. Recent reports further demonstrate the anticancer potential of anti-GAPDH strategies, prompting a reconsideration of GAPDH as a potential therapeutic target [20]. The preliminary correlation analysis using the PCAS suggests that *GAPDH* may also be regulated by m^6^A RNA methylation. This is particularly evident in lung adenocarcinoma and squamous cell carcinoma, where *GAPDH* expression shows the strongest correlation with the m^6^A reader protein IGF2BP1. Further experiments confirmed the presence of m^6^A modifications on *GAPDH* RNA, and the knockdown of IGF2BP1 reduced the stability of *GAPDH* RNA. These results validate the predictions made by the PCAS tool and highlight one of its application scenarios.

In conclusion, the PCAS package and Shinyapp significantly advance cancer research by leveraging the CPTAC database for comprehensive proteomic and phosphoproteomic analysis. The PCAS extends beyond basic data processing, incorporating advanced modules for detailed phosphorylation site analysis, immune cell infiltration, and drug sensitivity profiling. This holistic approach enhances our understanding of the tumor microenvironment and aids in developing targeted treatment strategies. By providing robust analytical tools for biomarker discovery and therapeutic predictions, the PCAS accelerates the transition from basic research to clinical application, setting a new benchmark for cancer research platforms.

## 4. Materials and Methods

### 4.1. Data Source

We downloaded multiple research cohort datasets of proteomics and phosphoproteomics data along with sample clinical information from the Proteomic Data Commons (PDC, https://pdc.cancer.gov/pdc/, accessed on 27 March 2024), and corresponding transcriptomic sequencing data from the Genomic Data Commons Data Portal (https://portal.gdc.cancer.gov/, accessed on 27 March 2024). For proteomics data, our current version only includes proteomics datasets based on the TMT method to ensure consistency in data standardization. Subsequent updates will include more proteomic datasets to enhance the 
analytical credibility of this tool. The log2 Ratio (only unshared peptides) data were used as the quantification metric for each protein. For the phosphoproteomics data, only log2 Ratio (All Peptides) data are provided on the PDC. The transcriptomics data were analyzed using log2 transformed data (TPM + 0.001). For samples that were analyzed more than once, we calculated the 
average of all replicates. In total, we downloaded and curated 23 proteomics 
datasets, 20 phosphoproteomics datasets, and 13 transcriptomics datasets. All 
clinical information of the samples was downloaded from the CPTAC database, and 
the OS time and status of each patient were calculated based on the provided 
follow-up time data. Details of each dataset can be found in Appendix A.

### 4.2. Software

This analysis tool was entirely developed using R language (Version 4.3.1), including the construction of a user visualization interface and analysis scripts. The main packages used and their primary functions are listed in Table 1.

### 4.3. Differential Expression Analysis

For the transcriptomic and proteomic data, we analyzed the differential expression of RNAs and proteins using the limma package [33] and *t*-tests. In addition, the *p*-value was corrected using the Benjamin and Hochberg (BH) method.

### 4.4. Immune Cell Infiltration Data

Based on transcriptomic data, we analyzed the immune cell infiltration scores of each sample using the “IOBR” package [34]. In this tool, we incorporated immune cell infiltration data derived from eight algorithms, including CIBERSORT [35], ESTIMATE [36], quanTIseq [37], TIMER [38], IPS, MCPCounter [39], xCell [40], and EPIC [41].

### 4.5. Drug Sensitivity Analysis

Using transcriptomic data, the “OncoPredict” package [8] was employed to analyze the sensitivity of each sample to 198 anti-cancer drugs, based on IC50 data in tumor cells and high-throughput sequencing data from tumor cells in the GDSC2.0 database [42]. Notably, the drug sensitivity scores calculated with this algorithm are positively correlated with IC50 values. Furthermore, these 198 drugs were categorized based on their target pathways, such as the cell cycle, WNT, and P53 signaling pathways, according to the annotated information from the GDSC2.0 database (https://www.cancerrxgene.org/, access data: 26 April 2024).

### 4.6. Using the Analysis Tool

We provide two ways for users to utilize the analysis tool. The first method is to access it online directly. We have deployed the Shiny application on the Shinyapp.io website. The link is as follows: https://jingle.shinyapps.io/PCAS/ accessed on 15 April 2024. Users can click the link to use the application directly in their browser for analysis.

The second method is to install the PCAS R package locally. Users can run the following command in their R environment to install the PCAS R package: remotes::install_github(“WangJin93/PCAS”). After installation, you can start the PCAS application locally by running the following command: PCAS::PCAS_app(). Additionally, the PCAS R package includes functions for data acquisition, analysis, and visualization, which can be used directly in the R terminal without relying on the Shinyapp.

### 4.7. Prediction of m^6^A Sites on GAPDH mRNA Using SRAMP

To identify potential m^6^A modification sites on GAPDH mRNA, the SRAMP prediction tool (http://www.cuilab.cn/sramp/, access data: 25 April 2024) was employed [43]. This machine learning-based platform predicts m^6^A sites based on nucleotide sequence motifs and known structural data, providing a probabilistic score for each site.

### 4.8. Cell Culture

The paper describes new software, but also mentions experiments; the following assays were used as an example of the validation of the findings performed with the tool. A549 and H1299 lung cancer cell lines, obtained from the American Type Culture Collection (ATCC), were cultured in RPMI 1640 medium, supplemented with 10% fetal bovine serum (FBS), 100 units/mL penicillin, and 100 µg/mL streptomycin in a humidified 5% CO_2_ atmosphere at 37 °C.

### 4.9. Construction of IGF2BP1 Knockdown shRNA Plasmid

IGF2BP1 gene silencing was achieved by designing specific shRNA sequences that target IGF2BP1 mRNA. These sequences were synthesized and cloned into the pGreen shRNA vector system. The resulting plasmids were transfected into cells using Lipofectamine 2000 according to the manufacturer’s protocol.

### 4.10. RNA Stability Assay

To assess the stability of *GAPDH* mRNA, cells were treated with 10 µg/mL actinomycin D, a transcription inhibitor, for specific time intervals. Samples were collected at 0, 1, 2, 3, 4, 5, and 6 h post-treatment. Total RNA was extracted using the TRIzol reagent, and the decay of GAPDH mRNA was quantified using real-time PCR. Ct values at different time points were normalized to the Ct value at t = 0 to obtain the ∆Ct value (∆Ct = average Ct of each time point—average Ct of t = 0), and the relative RNA abundance at each time point was calculated using the following formula: 2^−∆CT^. The mRNA decay rate was determined using a non-linear regression curve fitting (one phase decay). Primer sequence information can be found in Appendix A.

### 4.11. Quantitative PCR

The quantitative PCR (qPCR) analysis was performed using the SYBR Green method. Total RNA was extracted with Trizol and reverse transcribed into cDNA. qPCR was conducted on an ABI 7500 Real-Time PCR System with a total volume of 20 μL, containing SYBR Green Master Mix, forward primer, reverse primer, cDNA template, and nuclease-free water. The reaction conditions were as follows: 95 °C for 3 min, followed by 40 cycles of 95 °C for 15 s, 60 °C for 30 s, and 72 °C for 30 s. Each sample was run in triplicate. Relative gene expression levels were calculated using the 2^-ΔΔCt^ method and normalized to ACTB as an internal control.

### 4.12. Statistical Analysis

The statistical analysis of the data was performed using GraphPad V8.3.0 software (GraphPad Software, LLC). For comparisons between two groups, the Student’s *t*-test was applied. The significance level was set at *p* < 0.05. For RNA stability experiments, the decay of mRNA over time was analyzed using the one-phase decay model in GraphPad Prism. This model allowed us to estimate the half-life of mRNA by fitting the experimental data to an exponential decay function, where the rate constant derived from the fit corresponds to the decay rate of the mRNA. The results are expressed as mean ± SD, and differences were considered significant at *p* < 0.05.

## Figures and Tables

**Figure 1 ijms-25-06690-f001:**
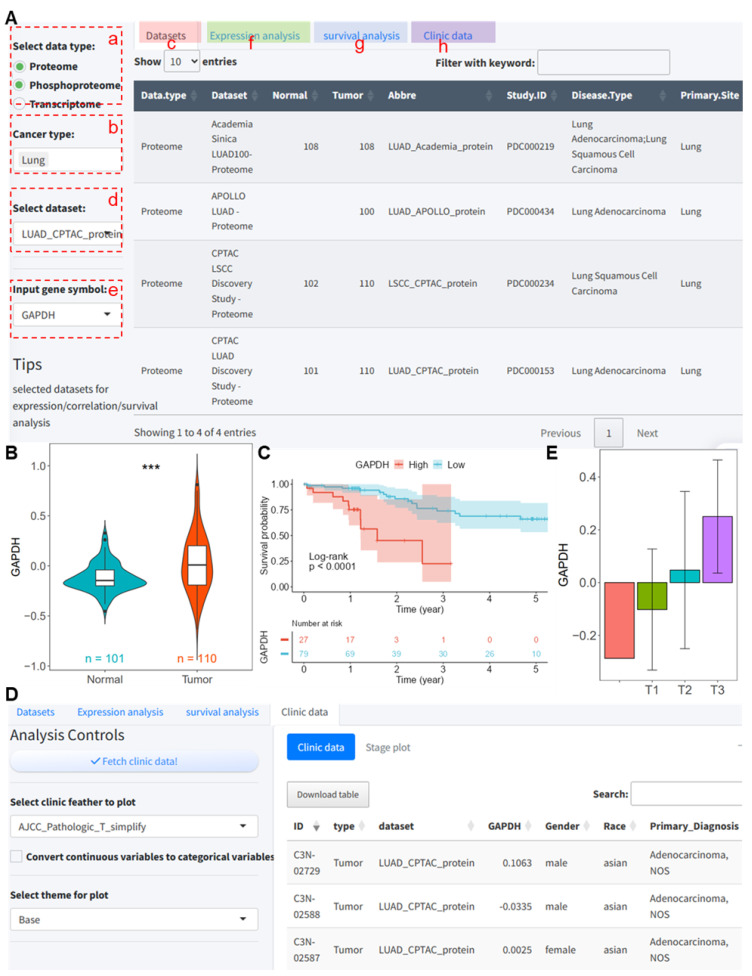
Analysis of GAPDH expression and prognosis in the LUAD_CPTAC_protein dataset using the analysis tool. (**A**) User interface demonstration; (**B**) differential protein expression of GAPDH in tumor and normal tissues in the LUAD_CPTAC_protein dataset; (**C**) survival curves for high and low GAPDH expression groups based on the LUAD_CPTAC_protein dataset; (**D**) clinical data retrieval in the analysis tool from the LUAD_CPTAC_protein dataset; and (**E**) bar plot showing differences in GAPDH expression across different tumor pathologic T stages. CPTAC, Clinical Proteomic Tumor Analysis Consortium. LUAD, Lung adenocarcinoma. ***, *p* < 0.001 between normal and tumor groups.

**Figure 2 ijms-25-06690-f002:**
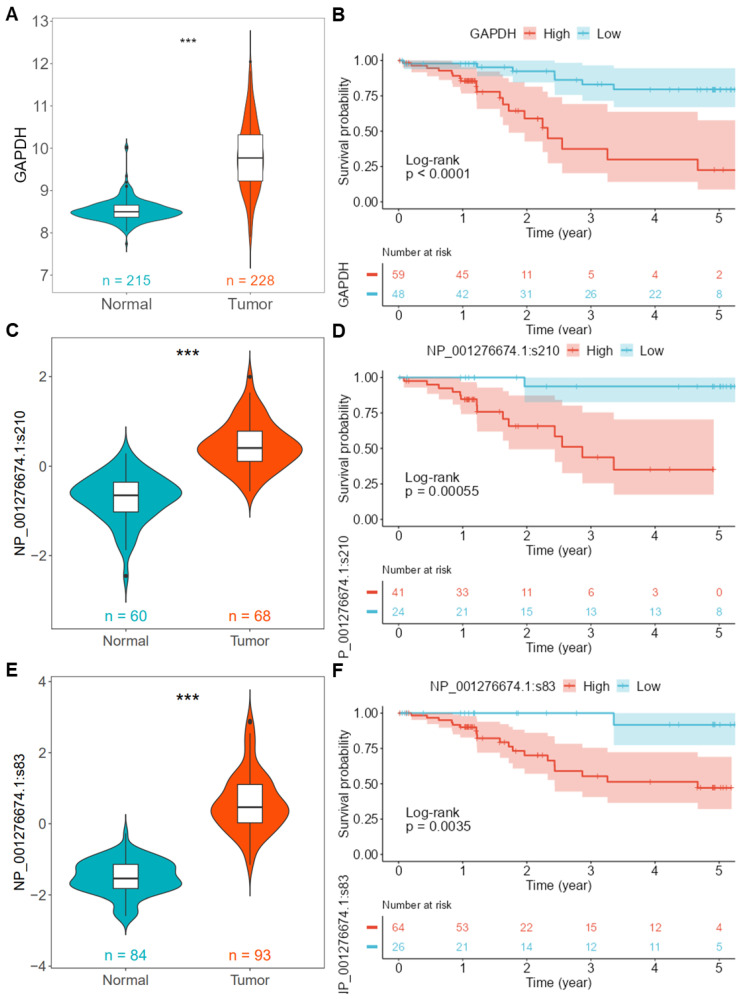
Analysis of *GAPDH* mRNA and phosphorylation site expression and prognosis using the PCAS tool. (**A**) Differential mRNA expression of *GAPDH* in tumor versus normal tissues in the LUAD_CPTAC_mRNA dataset. (**B**) Survival curves for high and low *GAPDH* expression groups based on the LUAD_CPTAC_mRNA dataset. (**C**) Differential expression of the GAPDH phosphorylation site NP001276674.1:s210 in tumor versus normal tissues in the LUAD_CPTAC_Phospho dataset. (**D**) Survival curves for high and low groups based on phosphorylation site NP001276674.1:s210 in the LUAD_CPTAC_Phospho dataset. (**E**) Differential expression of the GAPDH phosphorylation site NP001276674.1:s83 in tumor versus normal tissues in the LUAD_CPTAC_Phospho dataset. (**F**) Survival curves for high and low groups based on phosphorylation site NP001276674.1:s83 in the LUAD_CPTAC_Phospho dataset. ***, *p* < 0.001 between normal and tumor groups. CPTAC, Clinical Proteomic Tumor Analysis Consortium. LUAD, Lung adenocarcinoma.

**Figure 3 ijms-25-06690-f003:**
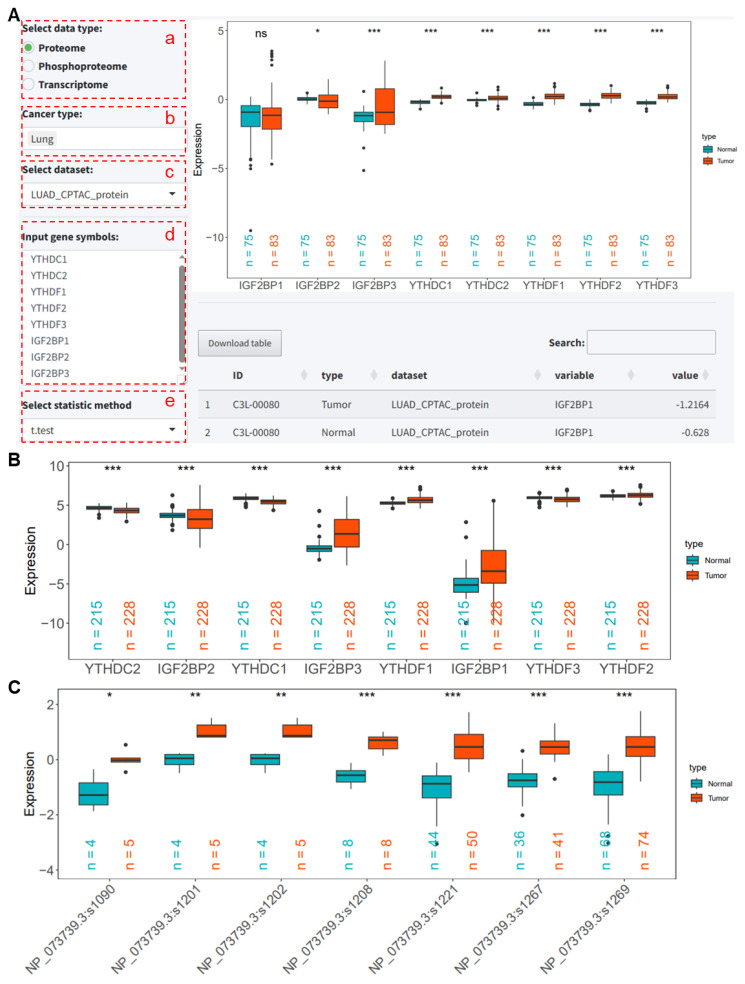
Analysis of m^6^A reader protein expression, RNA levels, and phosphorylation using the PCAS tool in the LUAD_CPTAC cohort. (**A**) PCAS tool interface for analyzing m^6^A reader protein levels in the LUAD_CPTAC_protein dataset. (**B**) mRNA levels of m^6^A reader proteins in the LUAD_CPTAC_mRNA dataset. (**C**) Phosphorylation levels of YTHDC2 phosphorylation sites in the LUAD_CPTAC_phoso dataset. *, *p* < 0.05 between two groups. **, *p* < 0.01 between two groups. ***, *p* < 0.001 between two groups. ns, not significant between two groups. CPTAC, Clinical Proteomic Tumor Analysis Consortium. LUAD, Lung adenocarcinoma.

**Figure 4 ijms-25-06690-f004:**
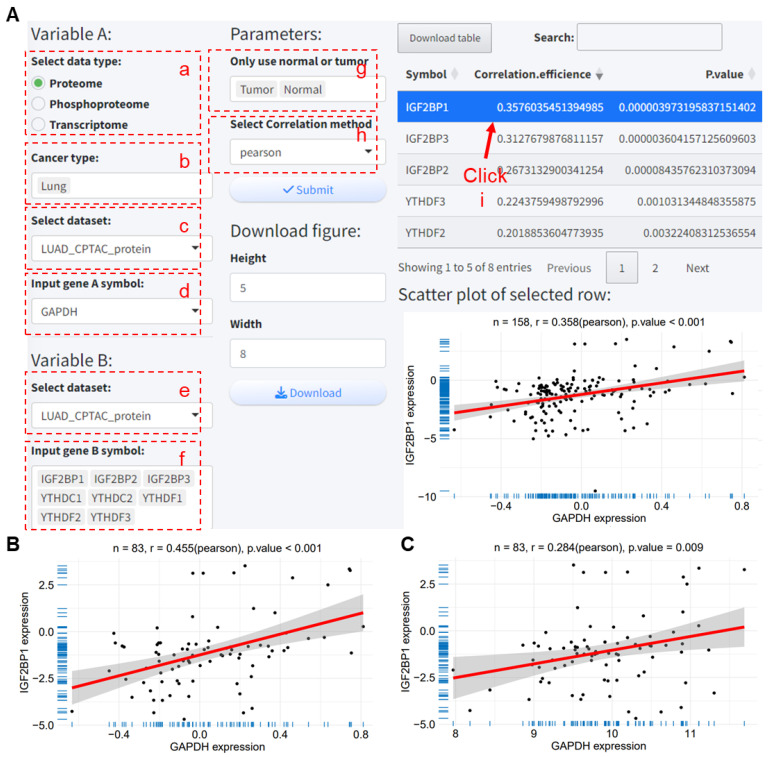
Analysis of the correlation between GAPDH and m^6^A reader protein expression in the LUAD_CPTAC_protein dataset using the PCAS tool. (**A**) PCAS tool interface for analyzing the correlation between GAPDH and m^6^A reader protein expression across all samples in the LUAD_CPTAC_protein dataset. (**B**) Correlation between GAPDH and IGF2BP1 protein expression in tumor samples from the LUAD_CPTAC_protein dataset. (**C**) Analysis of the correlation between *GAPDH* RNA expression and IGF2BP1 protein expression based on the LUAD_CPTAC_protein and LUAD_CPTAC_mRNA datasets. CPTAC, Clinical Proteomic Tumor Analysis Consortium. LUAD, Lung adenocarcinoma.

**Figure 5 ijms-25-06690-f005:**
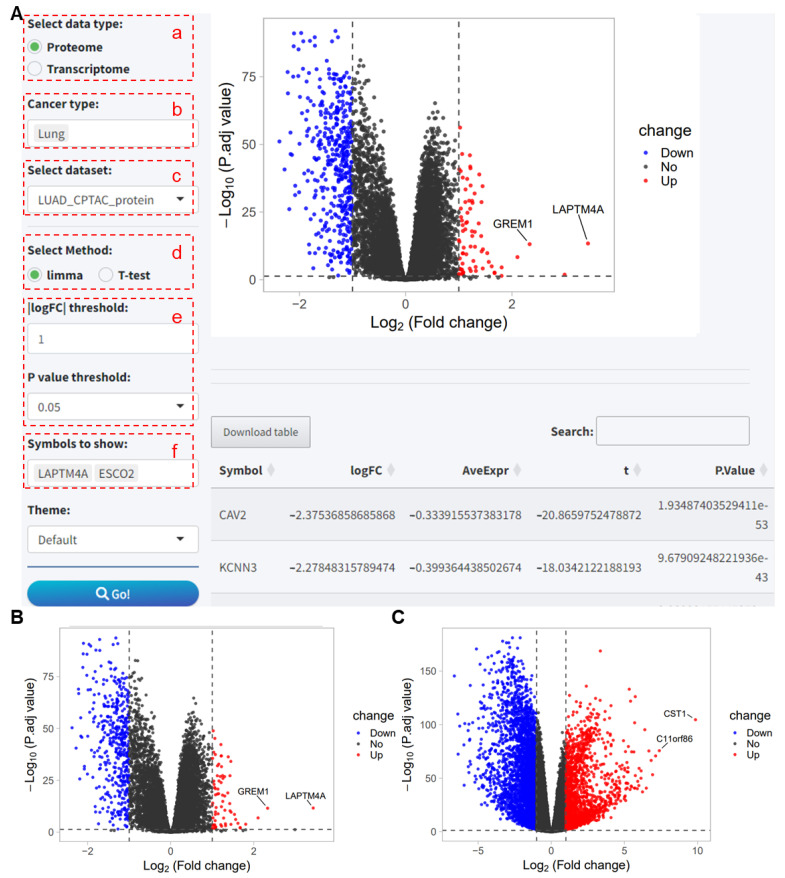
Differential gene expression analysis of proteomic and transcriptomic data from the CPTAC study cohort using the PCAS tool. (**A**) Differential protein expression analysis in the LUAD_CPTAC_protein dataset using the limma package, marking the two proteins with the highest fold changes. (**B**) Differential protein expression analysis in the LUAD_CPTAC_protein dataset using *t*-tests, marking the two proteins with the highest fold changes. (**C**) Differential gene expression analysis in the LUAD_CPTAC_mRNA dataset using limma, marking the two genes with the highest fold changes. CPTAC, Clinical Proteomic Tumor Analysis Consortium. LUAD, Lung adenocarcinoma.

**Figure 6 ijms-25-06690-f006:**
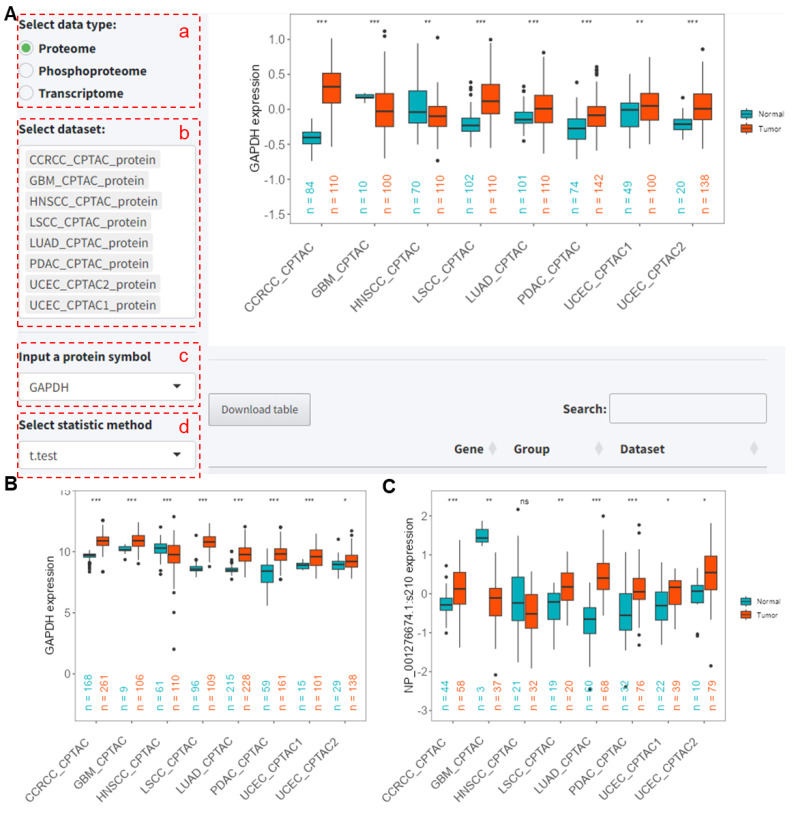
Analysis of *GAPDH* expression in proteins, RNA, and phosphorylation levels across multiple datasets. (**A**) PCAS interface for analyzing GAPDH protein expression levels across various datasets. (**B**) Differential RNA expression analysis of *GAPDH* across multiple transcriptomic datasets. (**C**) Differential analysis of the GAPDH phosphorylation site NP_001276674.1:s210 across multiple phosphoproteomic datasets. *, *p* < 0.05 between two groups. **, *p* < 0.01 between two groups. ***, *p* < 0.001 between two groups. ns, not significant between two groups. CCRCC: Clear Cell Renal Cell Carcinoma. GBM: Glioblastoma Multiforme. HNSCC: Head and Neck Squamous Cell Carcinoma. LSCC: Lung Squamous Cell Carcinoma. LUAD: Lung Adenocarcinoma. PDAC: Pancreatic Ductal Adenocarcinoma. UCEC: Uterine Corpus Endometrial Carcinoma. CPTAC, Clinical Proteomic Tumor Analysis Consortium.

**Figure 7 ijms-25-06690-f007:**
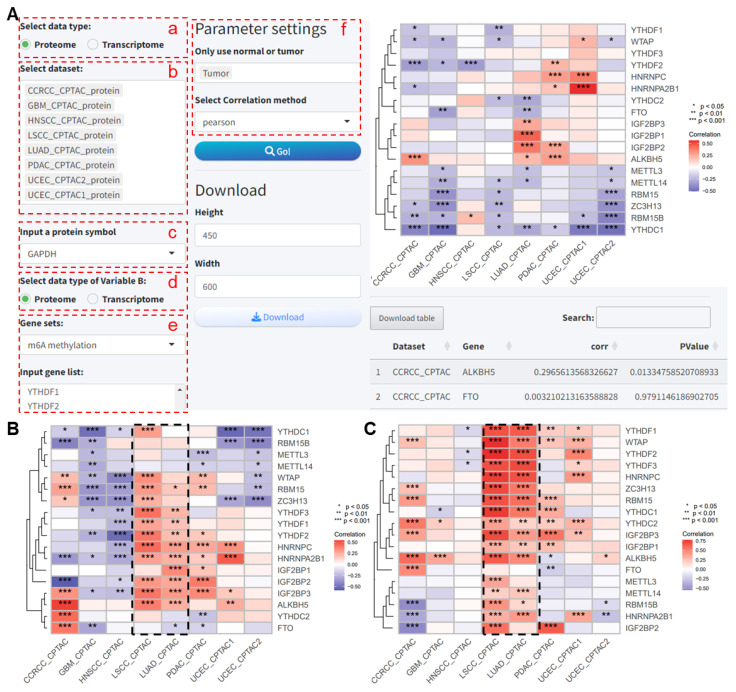
Analysis of the correlation between GAPDH and m^6^A regulatory proteins. (**A**) Analysis of the correlation between GAPDH and m^6^A regulatory proteins in tumor samples across multiple proteomic datasets. (**B**) Heatmap showing the correlation between GAPDH and m^6^A regulatory proteins across all samples in multiple proteomic datasets. (**C**) Analysis of the correlation between *GAPDH* mRNA expression and m^6^A regulatory proteins across multiple research cohorts. CCRCC: Clear Cell Renal Cell Carcinoma. GBM: Glioblastoma Multiforme. HNSCC: Head and Neck Squamous Cell Carcinoma. LSCC: Lung Squamous Cell Carcinoma. LUAD: Lung Adenocarcinoma. PDAC: Pancreatic Ductal Adenocarcinoma. UCEC: Uterine Corpus Endometrial Carcinoma. CPTAC, Clinical Proteomic Tumor Analysis Consortium.

**Figure 8 ijms-25-06690-f008:**
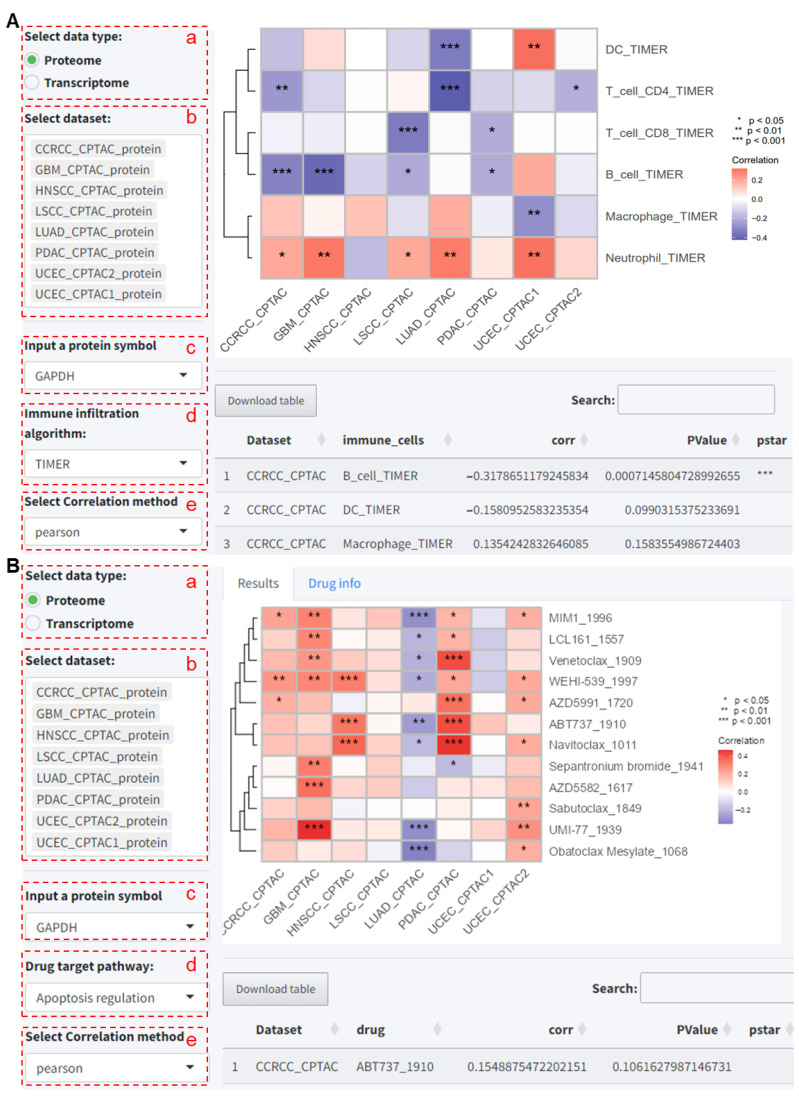
Analysis of the correlation between GAPDH and immune cell infiltration scores. (**A**) Analysis of the correlation between GAPDH and immune cell infiltration scores based on the TIMER algorithm across multiple proteomic datasets in tumor samples. (**B**) Analysis of the correlation between GAPDH and drug sensitivity scores targeting apoptosis in tumor samples across multiple proteomic datasets. CCRCC: Clear Cell Renal Cell Carcinoma. GBM: Glioblastoma Multiforme. HNSCC: Head and Neck Squamous Cell Carcinoma. LSCC: Lung Squamous Cell Carcinoma. LUAD: Lung Adenocarcinoma. PDAC: Pancreatic Ductal Adenocarcinoma. UCEC: Uterine Corpus Endometrial Carcinoma. CPTAC, Clinical Proteomic Tumor Analysis Consortium. TIMER, Tumor Immune Estimation Resource.

**Figure 9 ijms-25-06690-f009:**
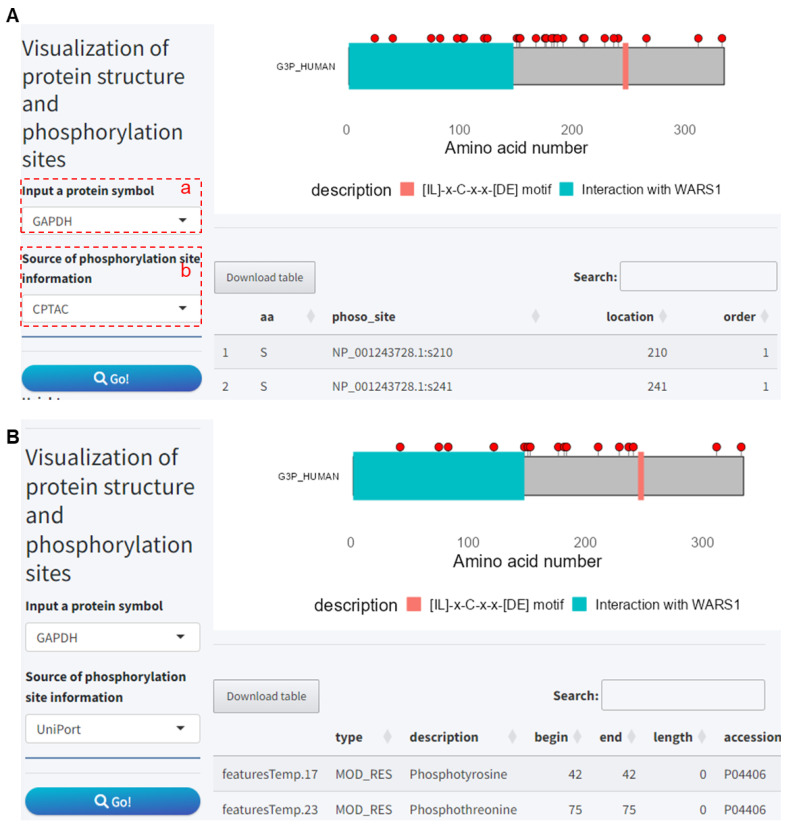
Visualization of protein structure and phosphorylation sites. (**A**) Visualization of phosphorylation site data obtained from the CPTAC proteomics database. (**B**) Visualization of phosphorylation site data obtained from the UniProt database. CPTAC, Clinical Proteomic Tumor Analysis Consortium.

**Figure 10 ijms-25-06690-f010:**
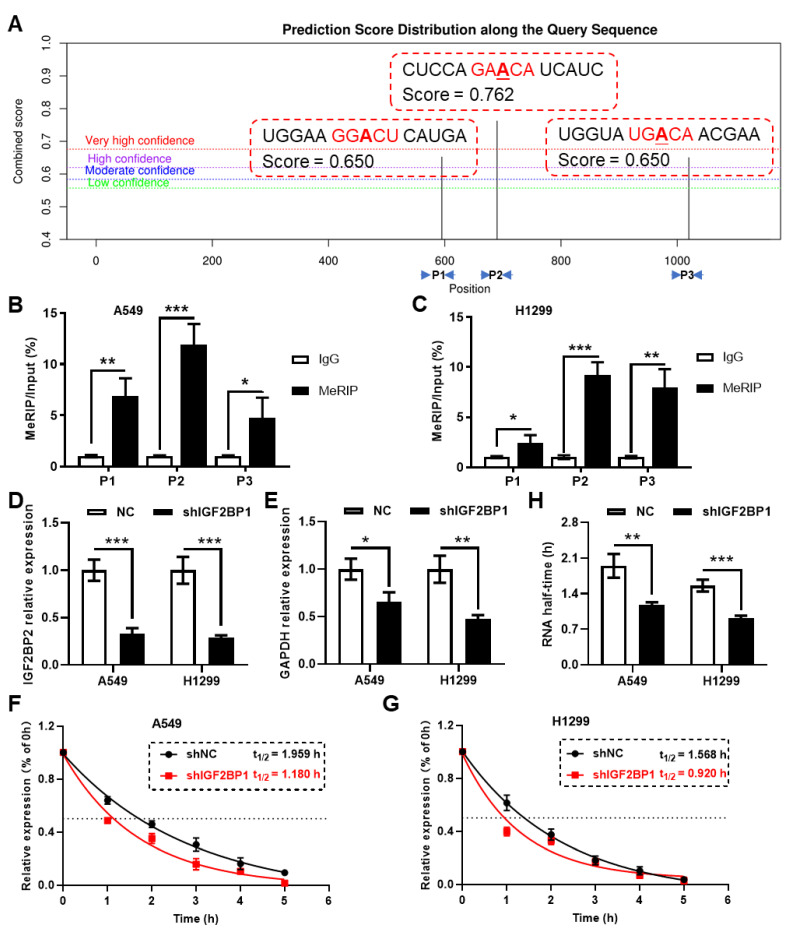
IGF2BP1 regulates the stability of GAPDH RNA. (**A**) Prediction of m^6^A modification sites in *GAPDH* mRNA using the SRAMP online tool. meRIP-qPCR analysis of the predicted m^6^A sites in A549 (**B**) and H1299 (**C**) cells. qPCR analysis of (**D**) *IGF2BP2* and (**E**) *GAPDH* expression changes in IGF2BP1 knockdown cells. RNA stability assays confirm changes in the half-life of *GAPDH* RNA following IGF2BP1 knockdown in (**F**) A549 and (**G**) H1299 cells. (**H**) The statistic results of half-time obtained from RNA stability assays. *, *p* < 0.05 between two groups. **, *p* < 0.01 between two groups. ***, *p* < 0.001 between two groups. meRIP, methyl-RNA immunoprecipitation.

**Table 1 ijms-25-06690-t001:** R packages used in PCAS tool.

R Package	Version	Functionality Description
shiny [21]	1.8.0	Framework for building interactive web applications directly from R.
bs4Dash [22]	2.3.0	A Bootstrap 4 shiny dashboard template for creating stylish dashboards.
shinyWidgets [23]	0.8.0	Enhances shiny by providing a variety of custom widgets, such as buttons, sliders, and more.
ggplot2 [24]	3.4.4	Data visualization
ggpubr [25]	0.6.0	Differential analysis of data between normal and tumor tissues
survminer [26]	0.4.9	Survival analysis and visualization
survival [27]	3.5.7	Survival analysis
dplyr [28]	1.1.4	A grammar of data manipulation, providing a consistent set of verbs that help you solve the most common data manipulation challenges.
psych [29]	2.3.12	Using the Corr. test function for correlation analysis
aplot [30]	0.2.2	Provides tools to decorate plots with a syntax that is coherent with ggplot2, and to create complex arrangements of plots.
ggtree [31]	3.8.2	An extension of ggplot2 to visualize phylogenetic trees with their annotations data and other associated data.
drawProteins [32]	1.20.0	Specifically designed for plotting protein schematics using ggplot2-based syntax.

## Data Availability

The datasets analyzed for this study are publicly available in the Clinical Proteomic Tumor Analysis Consortium (CPTAC) database. Access to the data can be obtained through the CPTAC data portal at https://proteomic.datacommons.cancer.gov/pdc/ (accessed on 27 March 2024), which offers open access to researchers following registration. The PCAS package source code has been published on Github: https://github.com/WangJin93/PCAS (accessed on 4 June 2024).

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
