# Peer review of "PCAS: An Integrated Tool for Multi-Dimensional Cancer Research Utilizing Clinical Proteomic Tumor Analysis Consortium Data"

_ijms, 2024, doi:10.3390/ijms25126690_

Round 1

Reviewer 1 Report

Comments and Suggestions for Authors

The manuscript authored by Jin Wang et al. describes the analysis of some CPTAC datasets. After reading the manuscript it is not entirely clear what is the main aim of authors. It is to describe a graphical user interface for researchers to re-analyze the datasets from CPTAC studies? Is it to find novel tumor biomarkers? It is to describe a step-by-step tutorial guide for the re-analysis of the CPTAC data? The authors suggest GAPDH and other proteins as novel tumor biomarkers, but it is not enough description of their analysis to be able to judge the validity of their claims. There is no comprehensive description of how the data was scaled, normalized, transformed or any other type of manipulation to reach such conclusions. Moreover, there are not comprehensive supplemental tables to sustain their conclusions. If the authors intend to describe a new tool available to research users to query the CPTAC datasets, then this should have been made available to the peer-reviewers. As such there is no way to rigorously review their results and conclusions. Considering all these and the specific items described below I cannot recommend publication.

Major:

1. How did the authors reached to GAPDH? Do they have an interest in this protein from previous studies, or are any other studies suggesting GAPDH as a tumor biomarker?

2. What kind of tumors do the authors refer to? Are specific cancers, related to a certain stage? Also, the “normal” group is not characterized: there are so many variables age, sex, smoking status etc. that could influence these outcomes. The information provided in the supplementary material is scarce. Do these GAPDH expression patterns are prevalent in some specific datasets? There is no comprehensive description of how the data was normalized/scaled/transformed between different datasets to make them comparable.

3. The data suggesting GAPDH as tumor biomarker is based on cell culture studies, fresh tissue, frozen tissue, FFPE etc? None of these are described.

4. Why they selected only TMT labeled experiments? Why not considering any other labeling procedures or label free data? This could create a method-specific bias for their conclusions.

5. Figure 1A: It appears there is a much wider expression level in the “Tumor” compared with Normal, rather than a higher expression. What statistical test was used to obtain the reported p-value? Was this adjusted? Such a wide spread of the data in the “Tumor” set would hardily result in a such a low p-value. Similar observations are valid for Figure 2A.

6. P2, L82-83: What are the optimal cutoff points? How were these selected? What do these denote? Are these optimal for what?

7. P2, L92: GAPDH can be phosphorylated to multiple sites. What was the rationale of selecting those particular phosphorylation sites?

8. Similar for the multigene expression analysis: Figure 3 suggests that the analyzed genes were pre-imputed as gene symbols to denote the differential analysis between tumor and non-tumor. How do the authors reach to this gene list?

Minor:

1. The a,b,c etc. description of various steps should be incorporated in the Materials and Methods section of the manuscript, not in the results, unless the authors intend to provide a tutorial guide to the CPTAC datasets analysis. This is also valid for the other sections of the manuscript mentioning such steps.

2. Figure 1C&2B: What do the red & blue rectangles in the background overlapping denote?

3. The p-values in the volcano plots should be adjusted according to the multiple testing procedure.

4. What is the difference between the correlation analysis from section 2.1.3 and 2.2.2?

5. The authors suggest that GAPDH expression is correlated with neutrophil, T cell, B cell infiltration and a higher sensitivity for multiple drugs. These should be validated by additional wet-lab experiments to draw such conclusions.

6. UniProt is a regularly updated database. The fact that the authors report additional phosphorylation sites for GAPDH which were found in already published data does not denote that these are sites which were not reported previously.

7. Figure 10: The level of IGF2BP1 is not shown for ctrl and KO cells. The half-life in time difference is significant? It should be tested for significancy.

8. The sequence of the primers used by the authors is not described.

Author Response

Dear professor,

Thank you for your comments concerning our manuscript entitled "ProteoCancer Analysis Suite (PCAS): An Integrated Platform for Multi-dimensional Cancer Research Using CPTAC Data" (ID: ijms-3017019). Those comments are all valuable and very helpful for revising and improving our paper, as well as providing important guidance for our research.

We have carefully studied the comments and made corrections, which we hope will meet with your approval. We would like to emphasize that the main purpose of this manuscript is to introduce the bioinformatics analysis tool, PCAS, which we have developed based on the multi-omics data from the CPTAC database. We chose to use GAPDH as a demonstration example because our previous research has suggested its potential as an oncogene in multiple cancers (GAPDH: A common housekeeping gene with an oncogenic role in pan-cancer. PMID: 37664172. DOI: 10.1016/j.csbj.2023.07.034.).

To improve the usability of the tool and the transparency of the backend analysis workflow, we have created an R package and uploaded it to GitHub. Users can install the R package and run the Shiny app locally.

Here are our point-by-point responses to the questions raised:

Major:

  1. How did the authors reached to GAPDH? Do they have an interest in this protein from previous studies, or are any other studies suggesting GAPDH as a tumor biomarker?

Reply: Thank you for your question. We acknowledge that our original manuscript did not disclose why GAPDH was chosen to demonstrate the application of our PCAS analysis platform. In fact, our previous study revealed the significant biological role of GAPDH in various cancers (GAPDH: A common housekeeping gene with an oncogenic role in pan-cancer. PMID: 37664172. DOI: 10.1016/j.csbj.2023.07.034). This study, based on bioinformatics analyses and in vitro experiments, found that GAPDH is significantly upregulated in many cancers and is associated with poor prognosis, immune cell infiltration, and immune checkpoint gene expression. Additionally, we predicted the upstream regulatory mechanisms of its abnormal expression and validated these predictions through functional and mechanistic in vitro experiments. In the current manuscript, we further analyze the protein level of GAPDH to illustrate its biological significance. We have supplemented the revised manuscript with relevant background and discussion.

  1. What kind of tumors do the authors refer to? Are specific cancers, related to a certain stage? Also, the “normal” group is not characterized: there are so many variables age, sex, smoking status etc. that could influence these outcomes. The information provided in the supplementary material is scarce. Do these GAPDH expression patterns are prevalent in some specific datasets? There is no comprehensive description of how the data was normalized/scaled/transformed between different datasets to make them comparable.

Reply: Thank you for your question. The primary goal of this manuscript is to introduce a bioinformatics analysis tool, PCAS, based on proteomics and transcriptomics data from the CPTAC database. All proteomics, transcriptomics data, and clinical sample information are downloaded from the CPTAC database. For proteomics data, we use the log2(Unshared peptide) values provided by the CPTAC database, which are already normalized and can be directly used for differential and correlation analyses. For transcriptomics data, we extract TPM values as gene expression levels and further perform log2 transformation (i.e., log2(TPM + 0.001)). We have detailed the data sources and normalization processes in the Methods section of the revised manuscript.

  1. The data suggesting GAPDH as tumor biomarker is based on cell culture studies, fresh tissue, frozen tissue, FFPE etc? None of these are described.

Reply: Thank you for your suggestion. As mentioned in our response to Comment 1, our previous research, based on extensive data from the TCGA and GEO databases, has already identified the potential of GAPDH as a tumor biomarker. In this manuscript, we use the PCAS tool to further analyze and explore the differential expression and biological significance of GAPDH at the protein level, aiming to highlight both the significance of GAPDH and the application value of the tool. We agree that in vivo and clinical sample validation is necessary to further substantiate these findings. In our previous study, we have already conducted some in vitro experiments on GAPDH’s function and mechanism. In future studies, we will continue to explore this gene's functions and regulatory pathways through extensive bioinformatics analysis and experimental validation.

  1. Why they selected only TMT labeled experiments? Why not considering any other labeling procedures or label free data? This could create a method-specific bias for their conclusions.

Reply: Thank you for your suggestion. In fact, most proteomics data in the CPTAC database are obtained using the TMT labeling method. We have also considered including proteomics data from other labeling methods. We examined the analysis tools provided on the CPTAC website, such as cProSite, and another online analysis tool, UALCAN, which both include only TMT-labeled proteomics datasets from CPTAC projects. Our current approach is to include all TMT-labeled proteomics datasets in the first version to ensure uniform data normalization. Future updates will incorporate more proteomics datasets from other methods to strengthen the tool's analytical reliability.

  1. Figure 1A: It appears there is a much wider expression level in the “Tumor” compared with Normal, rather than a higher expression. What statistical test was used to obtain the reported p-value? Was this adjusted? Such a wide spread of the data in the “Tumor” set would hardily result in a such a low p-value. Similar observations are valid for Figure 2A.

Reply: Thank you for your question. The differential analysis between Normal and Tumor samples shown in the manuscript was performed using a two-sample t-test, and the p-values were calculated using the ggpubr::compare_means() function with Benjamini & Hochberg correction for multiple comparisons (p.adjust.method = "BH"). The specific code has been published as part of the R package on GitHub. We have detailed this in the revised manuscript.

  1. P2, L82-83: What are the optimal cutoff points? How were these selected? What do these denote? Are these optimal for what?

Reply: Thank you for your question. The results shown in Figure 1 were obtained using the default parameters of our shiny app. For survival analysis, the default cutoff point mode is "Auto," determined by the survminer::surv_cutpoint function. Besides "Auto," users can also choose "Custom" to define their own cutoff points. We have elaborated on this in the Results section of the revised manuscript.

  1. P2, L92: GAPDH can be phosphorylated to multiple sites. What was the rationale of selecting those particular phosphorylation sites?

Reply: Thank you for your question. The selection of the two phosphorylation sites of GAPDH for analysis was merely for demonstrating the usage of our tool. The biological significance of these sites will be further validated in subsequent wet-lab experiments. We have clarified this in the revised manuscript.

  1. Similar for the multigene expression analysis: Figure 3 suggests that the analyzed genes were pre-imputed as gene symbols to denote the differential analysis between tumor and non-tumor. How do the authors reach to this gene list?

Reply: Thank you for your question. We apologize for any confusion caused by our unclear description. The genes used for the demonstration of this module are related to m6A RNA methylation regulation. We selected these genes because we found predicted methylation sites on GAPDH RNA, as shown in Figure 10A. We have clarified this in the revised manuscript.

Minor:

  1. The a,b,c etc. description of various steps should be incorporated in the Materials and Methods section of the manuscript, not in the results, unless the authors intend to provide a tutorial guide to the CPTAC datasets analysis. This is also valid for the other sections of the manuscript mentioning such steps.

Reply: Thank you for your suggestion. We apologize for not clearly stating the purpose of our manuscript. As you mentioned, our intent is to provide a tutorial guide for the CPTAC datasets analysis using our developed tool. The updated manuscript now clearly outlines this purpose.

  1. Figure 1C&2B: What do the red & blue rectangles in the background overlapping denote?

Reply: The red and blue rectangles appearing in the survival curve plot represent the confidence intervals of the curves, which indicate the uncertainty of the survival probability at each time point. This is because the limited sample size involved in the survival analysis leads to a certain degree of fluctuation in the survival probability at each time point. By observing the confidence intervals of the survival curves for the two groups, we can intuitively judge the significance of the difference in survival probability between the two groups. When the confidence intervals of the two groups overlap, it means that there is no statistically significant difference in the survival probability at that time point. When the confidence intervals of the two groups do not overlap, it indicates that there is a significant difference in the survival probability between the two groups at that time point.

  1. The p-values in the volcano plots should be adjusted according to the multiple testing procedure.

Reply: Thank you for your suggestion. We have modified the p-values in the volcano plot module of our shiny app to be adjusted using the Benjamini & Hochberg correction method.

  1. What is the difference between the correlation analysis from section 2.1.3 and 2.2.2?

Reply: The correlation analysis in section 2.1.3 is a module under the single dataset analysis framework, aiming to analyze the correlation between the target gene/protein and multiple genes/proteins within a single dataset. In contrast, section 2.2.2 is a module under the multi-dataset analysis framework, aiming to perform correlation analysis across multiple datasets or pan-cancer.

  1. The authors suggest that GAPDH expression is correlated with neutrophil, T cell, B cell infiltration and a higher sensitivity for multiple drugs. These should be validated by additional wet-lab experiments to draw such conclusions.

Reply: Thank you for your suggestion. The correlation analysis between GAPDH expression and immune cell infiltration and drug sensitivity aims to demonstrate the usage of our bioinformatics analysis tool. The R packages used for calculating immune cell infiltration and drug sensitivity are widely accepted. Our previous study, based on the TCGA database, also found significant correlations between GAPDH transcription levels and immune cell infiltration and drug sensitivity. This study based on the CPTAC database serves as a valid supplement. We plan to validate these bioinformatics analysis results through in vivo and in vitro experiments in our future studies.

  1. UniProt is a regularly updated database. The fact that the authors report additional phosphorylation sites for GAPDH which were found in already published data does not denote that these are sites which were not reported previously.

Reply: Thank you for your question. Our study does not report additional phosphorylation sites for GAPDH. The phosphorylation sites mentioned in this manuscript are annotated in the CPTAC phosphoproteomics data. The role of the UniProt database in this manuscript is to obtain protein sequence structure data for visualizing phosphorylation sites.

  1. Figure 10: The level of IGF2BP1 is not shown for ctrl and KO cells. The half-life in time difference is significant? It should be tested for significancy.

Reply: Thank you for your suggestion. We have added the results for IGF2BP1 levels in ctrl and KO cells and tested the significance of the half-life time difference in the revised manuscript.

  1. The sequence of the primers used by the authors is not described.

Reply: Thank you for your suggestion. We have added the primer information in the revised manuscript as shown in Table S2.

We appreciate your valuable feedback, which has helped us improve the clarity and quality of our manuscript. Thank you again for your thorough review and helpful suggestions.

Reviewer 2 Report

Comments and Suggestions for Authors

ProteoCancer Analysis Suite (PCAS): An Integrated Platform for Multi-dimensional Cancer Research Using CPTAC Data

The authors present an analysis of data related to cancer research; the manuscript is well written to a high standard, but I was left confused about what was novel about the work, what actual methods were used, and how reproducible it was. Comments follow.

ABSTRACT

(1) The Abstract should do a better job of telling the reader what is NEW about the research. Is this a new workflow / platform that is presented for the first time? Are the results of GAPDH / FOXM1 new? What is the target audience for this manuscript?

INTRODUCTION

(2) Line 34, peptides are not mixed and then analysed in mass spectrometry experiments, they are usually separated then analysed, using liquid-chromatography then mass spectrometry. See PMID: 38287260 for a recent review, or others that the authors could usefully cite

(3) Line 39, change to "...it has been reported..."

(4) Line 41, change to "...the remaining alterations may be due to..."

(5) Line 51, PCAS, is this a new platform that the authors present for the first time? If the first time, make this clear. If previously presented, reference.

RESULTS

(6) I was unable to replicate the results as the code / platform is not referenced. The figures are, however, produced to a good standard in my view

DISCUSSION

(7) Same comments as Introduction and Abstract. I could not tell what the NEW / NOVEL contribution of the authors was. For example, GAPDH and FOXM1 are all well-known as cancer-related, see PMID: 30208972, PMID: 35931301, PMID: 23103567, which are all more helpful articles to cite than the single reference on GAPDH [16] given. The Discussion should include many more references to current literature so that the reader can put these findings into their wider context.

(8) Also, if the main 'new' aspect to this research is the PCAS platform, what can PCAS do that existing platforms used by other researchers cannot?

METHODS

(9) 4.7 Cell culture (line 419), I am confused, are the data analysed in this work downloaded from a repository, or did the authors do cell culturing themselves?

(10) Throughout the manuscript, where p-values are given, the authors should disclose whether these are FDR corrected when appropriate

DATA AND CODE AVAILABILITY

(11) Where will the code be available, Github, web service, Zenodo?

(12) Line 442, what supporting information, the authors do not specify this?

Comments on the Quality of English Language

English language is of a good standard

Author Response

Dear professor,

Thank you for your comments concerning our manuscript entitled "ProteoCancer Analysis Suite (PCAS): An Integrated Platform for Multi-dimensional Cancer Research Using CPTAC Data" (ID: ijms-3017019). Those comments are all valuable and very helpful for revising and improving our paper, as well as providing important guidance for our research.

We have carefully studied the comments and made corrections, which we hope will meet with your approval. We would like to emphasize that the main purpose of this manuscript is to introduce the bioinformatics analysis tool, PCAS, which we have developed based on the multi-omics data from the CPTAC database. We chose to use GAPDH as a demonstration example because our previous research has suggested its potential as an oncogene in multiple cancers (GAPDH: A common housekeeping gene with an oncogenic role in pan-cancer. PMID: 37664172. DOI: 10.1016/j.csbj.2023.07.034.).

To improve the usability of the tool and the transparency of the backend analysis workflow, we have created an R package and uploaded it to GitHub. Users can install the R package and run the Shiny app locally.

Here are our point-by-point responses to the questions raised:

ABSTRACT

(1) The Abstract should do a better job of telling the reader what is NEW about the research. Is this a new workflow / platform that is presented for the first time? Are the results of GAPDH / FOXM1 new? What is the target audience for this manuscript?

Reply: Thank you for your suggestion. We have reorganized the abstract to emphasize the value of our newly developed bioinformatics analysis tool presented in this manuscript.

INTRODUCTION

(2) Line 34, peptides are not mixed and then analysed in mass spectrometry experiments, they are usually separated then analysed, using liquid-chromatography then mass spectrometry. See PMID: 38287260 for a recent review, or others that the authors could usefully cite

Reply: We apologize for any confusion. In TMT-labeled proteomics analysis, peptides labeled with different TMT tags are indeed mixed together before being analyzed by liquid chromatography and then mass spectrometry.

(3) Line 39, change to "...it has been reported..."

Reply: Thank you for your careful review. The grammatical error has been corrected.

(4) Line 41, change to "...the remaining alterations may be due to..."

Reply: Thank you. We have made the suggested correction.

(5) Line 51, PCAS, is this a new platform that the authors present for the first time? If the first time, make this clear. If previously presented, reference.

Reply: Thank you for your question. The purpose of this manuscript is to share a newly developed multi-omics analysis tool, PCAS, based on CPTAC proteomics and transcriptomics data. We have revised the abstract and introduction to clearly highlight the purpose and significance of this manuscript.

RESULTS

(6) I was unable to replicate the results as the code / platform is not referenced. The figures are, however, produced to a good standard in my view

Reply: We apologize for any confusion caused by our oversight. As mentioned in our previous response, we have packaged the analysis tool as an R package and released it on GitHub. Users can perform the data analysis directly in R or run the PCAS app for a graphical interface. The revised manuscript includes the relevant links.

DISCUSSION

(7) Same comments as Introduction and Abstract. I could not tell what the NEW / NOVEL contribution of the authors was. For example, GAPDH and FOXM1 are all well-known as cancer-related, see PMID: 30208972, PMID: 35931301, PMID: 23103567, which are all more helpful articles to cite than the single reference on GAPDH [16] given. The Discussion should include many more references to current literature so that the reader can put these findings into their wider context.

Reply: Thank you for your suggestion. The main purpose of this manuscript is to introduce our developed bioinformatics analysis tool, with GAPDH chosen for demonstration due to our prior research on this gene. The discussion of FOXM1 is based on prior findings that it can regulate GAPDH. We have revised the manuscript to include more references and to clearly highlight the novelty and significance of the PCAS platform.

(8) Also, if the main 'new' aspect to this research is the PCAS platform, what can PCAS do that existing platforms used by other researchers cannot?

Reply: Thank you for your question. Currently, there are few analysis tools based on CPTAC data, such as UALCAN and cProSite. We have discussed the unique capabilities of PCAS compared to existing platforms in the first paragraph of the discussion section.

METHODS

(9) 4.7 Cell culture (line 419), I am confused, are the data analysed in this work downloaded from a repository, or did the authors do cell culturing themselves?

Reply: Thank you for your question. The source data for the bioinformatics analysis was indeed downloaded from the CPTAC database. The purpose of this manuscript is to introduce our newly developed bioinformatics analysis platform. We used the GAPDH gene, which we previously researched, as an example to demonstrate the tool's workflow. To further validate the reliability of the analysis results from the tool, we conducted necessary in vitro experiments ourselves. We have clarified this in the revised manuscript.

(10) Throughout the manuscript, where p-values are given, the authors should disclose whether these are FDR corrected when appropriate

Reply: Thank you for pointing this out. In our tool, all p-values related to gene expression differential analysis have been adjusted using the Benjamini & Hochberg method. We have made this clear in the revised manuscript.

DATA AND CODE AVAILABILITY

(11) Where will the code be available, Github, web service, Zenodo?

Reply: Thank you for your question. As mentioned in our previous response, we have packaged the analysis tool as an R package and released it on GitHub (https://github.com/WangJin93/PCAS). Users can perform the data analysis directly in R or run the PCAS app for a graphical interface. We have included the relevant links in the revised manuscript.

(12) Line 442, what supporting information, the authors do not specify this?

Reply: We apologize for the oversight. The revised manuscript now includes detailed information about the supporting materials and how they can be accessed.

We appreciate your valuable feedback, which has helped us improve the clarity and quality of our manuscript. Thank you again for your thorough review and helpful suggestions.

Round 2

Reviewer 2 Report

Comments and Suggestions for Authors

The authors have comprehensively and thoughtfully responded to my comments. I thank them for their efforts - the manuscript is much improved and in my view is suitable for publication.

Author Response

Dear professor,

We sincerely appreciate your positive feedback and kind words regarding our manuscript. We are grateful for your comprehensive and thoughtful comments, which have significantly contributed to improving the quality of our work.